# Living with Overweight, Rather than a History of Gestational Diabetes, Influences Dietary Quality and Physical Activity during Pregnancy

**DOI:** 10.3390/nu14030651

**Published:** 2022-02-03

**Authors:** Ella Muhli, Ella Koivuniemi, Kirsi Laitinen

**Affiliations:** 1Research Centre for Integrative Physiology and Pharmacology, Institute of Biomedicine, University of Turku, 20014 Turku, Finland; elromu@utu.fi (E.M.); elmkoi@utu.fi (E.K.); 2Department of Obstetrics and Gynecology, Turku University Hospital, 20521 Turku, Finland

**Keywords:** diet, physical activity, pregnancy, overweight, obesity, gestational diabetes

## Abstract

(1) Background: Clinical practice guidelines recommend dietary and physical activity counselling for pregnant women with gestational diabetes (GDM). The aim of this study was to evaluate the extent to which a history of GDM and living with overweight before pregnancy modify dietary quality and physical activity during pregnancy. (2) Methods: The study is a cross-sectional study of 1034 pregnant women from different parts of Finland. The data were collected through electronic questionnaires. Dietary quality and physical activity were measured with stand-alone indices and compared according to the history of GDM and overweight status based on body mass index (BMI) category. (3) Results: Overall, 53% of the women had a poor dietary quality (Index of Diet Quality (IDQ) score < 10) and 45% a light physical activity level. The IDQ score or physical activity levels did not differ between women with and without a history of GDM. Instead, in women with overweight/obesity both the IDQ score and physical activity levels were lower compared to their normal-weight counterparts (*p* < 0.001). (4) Conclusions: Pregnant women, particularly if living with overweight, commonly have a poor dietary quality and a light level of physical activity. A history of GDM is not reflected in the lifestyle habits, despite the assumption that they have received lifestyle counselling during a previous pregnancy. Pregnant women would benefit from new means to promote healthy lifestyle changes.

## 1. Introduction

There is an alarming trend that more and more pregnant women are living with overweight as this condition predisposes both mother and child to health complications, including an increased risk of gestational diabetes (GDM) [1]. GDM refers to diabetes first manifesting during pregnancy [2], currently diagnosed in approximately 15% of women [3]. In Finland, where this study was conducted, the value may be as high as one in every five [4]. Not only does GDM increase the risks of serious perinatal complications in the offspring [5], but it also may lead to serious health complications like type 2 diabetes after the pregnancy [6,7].

Conventional strategies to promote the pregnant woman’s health include counselling on dietary and physical activity habits [8,9] and regular maternity care visits which represent an excellent opportunity to provide that vital information. In Finland, the standard prenatal care for all women includes at least nine visits for primiparas and at least eight visits for multiparas. In line with the current care guidelines [10], women with GDM receive diet and exercise counselling from maternity care nurses in communal health centers. The aims of the counselling are maintaining normoglycemia and preventing excessive gestational weight gain by providing practical advice. In most cases, GDM can be controlled with lifestyle modifications alone [11]. While one might expect that pregnant women with a history of GDM will have improved their dietary quality as they have previously received dietary counselling, in fact, up till now the dietary quality and physical activity of pregnant women with a history of GDM have not been evaluated.

A good dietary quality signifies an adherence to a health-promoting diet as defined in dietary recommendations with a high consumption of vegetables, fruits and berries, whole grain foods, vegetable oil-based spread and low-fat dairy, together with a low consumption of sugary beverages and food high in saturated fats. A higher dietary quality has been associated with a lower risk of obesity in both pregnant and non-pregnant adults [12,13]. Because of this we hypothesized that the dietary quality of women with overweight or obesity when they are pregnant would deviate from that recommended. We aimed to evaluate the overall quality of the diet with a stand-alone validated index, specifically designed to reflect dietary recommendations [14].

The first aim of the present study was to determine the dietary quality and physical activity with stand-alone indices [14,15] in a sample from different parts of Finland of pregnant women who had been diagnosed with GDM in a previous pregnancy compared to those without history of GDM. The second aim was to evaluate the dietary quality and physical activity in women with overweight or obesity as compared to women with normal weight during pregnancy.

## 2. Materials and Methods

### 2.1. Study Design

Women were recruited into a study investigating lifestyle and the use of a mobile application during pregnancy by announcements in social media (Facebook) running from June to October 2017. The data at the baseline of the study are reported here in a cross-sectional design. We wanted to recruit a cohort of women in early to mid-pregnancy, so the criteria for inclusion were less than gestational week 28 and Finnish language skills. The women interested in the study (*n* = 1512) contacted the researcher via an electronic form and those who were eligible, were subsequently provided with further information about the study and sent the study questionnaires by e-mail. A total of 1047 women participated in the study. Ten questionnaires were excluded based on missing information on gestational weeks or gestational weeks ≥ 28 and 3 questionnaires were excluded based on no weight and/or height reported.

Data on dietary quality, physical activity, use of dietary supplements, current health status, height and weight were collected through electronic questionnaires. With respect to diet and physical activity, the data were collected at a mean week of 14.9 (standard deviation (SD) 6.4) during pregnancy. The diagnosis of prior GDM was self-reported. Based on the Finnish guidelines [10], nearly all pregnant women undergo a 2-h oral glucose tolerance test between gestational weeks 24 and 28. A diagnosis is made if blood glucose is at or above the threshold levels in one or more timepoints: 0 h 5.3 mmol/L, 1 h 10.0 mmol/L, 2 h 8.6 mmol/L. When comparing the study cohort to the population of pregnant women in Finland [4], we also collected data on locality, age, parity, marital status, education, household income (low income defined as less than 20,000 euros/year based on an income less than 60 percent of the median household income in Finland [16]), previous health conditions, smoking status, and adherence to special diets.

### 2.2. Measures

Dietary quality was inquired by a validated Index of Diet Quality (IDQ) designed to reflect the Nordic and Finnish nutrition recommendations [14]. The IDQ is comprised of 18 multiple-choice questions about the frequency and quality of consuming foods. The total IDQ score ranges from 0 to 15 points such that scores ≥ 10 are considered as health-promoting (good dietary quality), thus a score < 10 is designated as a poor dietary quality. Some components (vegetables, fruit and berries, whole-grain products, vegetable oil -based spread, fish, and meal pattern) of the IDQ were also individually compared to dietary recommendations as categorized variables (yes/no).

Leisure-time physical activity (LTPA) was measured using the metabolic equivalent (MET)-index which consists of three multiple-choice questions inquiring about the intensity, frequency, and duration of physical activity in leisure-time [15]. The total score ranges from 0 to 105 MET h/week (h/wk). The index was further categorized to light LTPA (<5 MET h/wk), moderate LTPA (≥5 but ≤30 MET h/wk), and vigorous LTPA (>30 MET h/wk). Level 5 MET h/wk corresponds to about one hour of moderate-intensity physical activity weekly and level 30 MET h/wk to about one hour of moderate-intensity physical activity every day [15].

### 2.3. Data Analysis

In the analyses, the participating women were assigned to either a group with normal weight (body mass index (BMI) < 25.0 kg/m^2^) or a group with overweight/obesity (BMI ≥ 25.0 kg/m^2^) based on their pre-pregnancy BMI. The women with underweight (*n* = 22) were included in the normal-weight group since they did not differ significantly in terms of either their IDQ score or MET-index. Primiparous women were excluded from the analyses when the women with a history of GDM were compared to the women who had not been diagnosed with this condition in their previous pregnancy.

No imputations regarding missing data were conducted. The normality of the data was observed from histograms and homogeneity of variances checked with Levene’s test. Categorical data are presented as frequencies and percentages and normally distributed continuous variables as means and standard deviations. As MET-indices were not normally distributed, these data are summarized as medians and interquartile ranges.

Comparisons for categorical data were made with Fisher’s exact test or the Chi-square test. MET-indices were compared using the Mann–Whitney U-test or the Kruskal–Wallis test. Independent samples T-test or one-way ANOVA was used to compare age, gestational weeks, pre-pregnancy BMI and IDQ scores between groups. When the women were divided into subgroups based on a history of GDM and pre-pregnancy BMI, a post hoc Tukey test was used to compare the IDQ scores.

Possible confounding factors were considered, and models of the IDQ score adjusted for age and/or pre-pregnancy BMI were made with ANCOVA. Pre-pregnancy BMI, university education and pre-pregnancy smoking status intercorrelated strongly and therefore the model was adjusted with pre-pregnancy BMI only.

Two-sided *p*-values <0.05 were considered significant. Analyses were executed with IBM SPSS statistics version 25.0 for Windows (IBM SPSS Inc., Chicago, IL, USA).

## 3. Results

### 3.1. Clinical Characteristics

The study population consisted of 1034 pregnant women from different parts of Finland (Appendix A). As indicated in Table 1, 37% of the women were living with overweight (23%) or obesity (14%). Of the multipara women, 18% reported having been diagnosed with GDM in a previous pregnancy, which is in accordance with the value in the general population (19%) [4].

The women with and without a history of GDM did not differ significantly with respect to mean age, parity, marital status, university education, or income (Table 1). However, the women with a history of GDM had a higher pre-pregnancy BMI, and they had smoked more commonly before pregnancy. No significant differences were evident between the characteristics of the women with normal weight and the women with overweight/obesity, except that the women with overweight/obesity had less frequently a university-level education and had smoked more commonly before pregnancy.

### 3.2. Dietary Quality and Physical Activity According to the History of GDM Status

Overall, 53% of the women had a poor dietary quality, but as shown in Table 2, the IDQ scores did not differ between the women with and without a history of GDM (mean 9.2 (SD 2.2) and mean 9.4 (SD 2.2), respectively), not even after adjusting for pre-pregnancy BMI. Furthermore, there were no significant differences in the individual components of the IDQ score between the groups (Table 3). When compared with the women with a history of GDM, those without were more likely to consume whole-grain products on a daily basis (*p* = 0.062). There were no significant differences in the frequency of using dietary supplements between the women with and without a history of GDM (Appendix A).

Most of the women were undertaking either light or moderate physical activity; 52% of those with a history of GDM were undertaking light physical activity (Table 2), but the physical activity levels or the MET-index did not differ between the groups.

### 3.3. Dietary Quality and Physical Activity According to the Overweight Status

Having a good dietary quality was significantly more common in the women with normal weight than in their counterparts with overweight/obesity (Table 4). The IDQ score of the women with normal weight (mean 9.6 (SD 2.0)) was also higher than that of the women with overweight/obesity (mean 8.9 (SD 2.3)). When evaluating the individual components of the IDQ score, women with overweight/obesity were less likely to be eating vegetables, fruit or berries, and whole-grain products on a daily basis (Table 3). In addition, they were less likely to consume at least five portions of vegetables, fruit or berries daily and more likely to have an irregular meal pattern. In the evaluation of the frequency of using food supplements, the women with overweight/obesity more likely used multivitamin supplements regularly than the women with normal weight but were less likely to consume any probiotics (Appendix A).

The women with overweight/obesity had more commonly a light physical activity level than the women with normal weight (Table 4). In addition, the women with overweight/obesity had a significantly lower median MET-index.

When we divided the women into subgroups based on both living with overweight/obesity and a history of GDM, the normal-weight women without a history of GDM had a significantly higher IDQ score than the women with overweight/obesity who had not experienced GDM in their past pregnancy (Appendix A, post hoc Tukey test, *p* < 0.001). The normal-weight groups had more commonly a good dietary quality than the groups with overweight/obesity, regardless of their GDM status. No significant differences were evident in physical activity.

## 4. Discussion

We demonstrated that approximately only half of the Finnish pregnant women studied here had a good overall dietary quality, and furthermore, in early to mid-pregnancy, a light level of physical activity was evident in almost one out of every two. Based on our results, the group of women most at risk for experiencing a pregnancy-related complication, i.e., the women with a history of GDM, did not differ from those without a history of GDM in terms of dietary quality or physical activity. It was also more common that the women with overweight/obesity consumed a poor-quality diet, as evaluated by the IDQ, and had a lower physical activity level during pregnancy than the women with normal weight.

A novel finding in our study was that a low dietary quality score and a light physical activity level were commonly encountered during pregnancy in the women with a history of GDM. This is worrying as these individuals carry a high risk for recurrent GDM [17]. In a previous study examining dietary quality within 10 years following a pregnancy affected with GDM, it was noted that the women with a history of GDM had a lower dietary quality score than those with no such history [18]. Women with a history of GDM have also been reported still to have a lower intensity of physical activity four years after pregnancy [19]. These findings, when combined with our new data, highlight the importance of focusing interventions on this risk population of pregnant women. The possible reasons for the unhealthy dietary and exercise habits in pregnant women with a history of GDM could be either a lack of knowledge or poor motivation. It is also possible that there is a lack of awareness among health care professionals of this group of mothers, who would need targeted dietary and exercise counselling as counselling practices in the communal health centers in Finland are known to be somewhat variable [20]. It is recommended that the dietary counselling for women with GDM should be tailored to the individual, practical, and given repeatedly [10], but it is possible that these are not sufficiently implemented due to limited resources. Increasing healthy eating and physical activity may manifest as clinically improved health as demonstrated in a trial where the risk of GDM was reduced by combined dietary and physical activity counselling given above the standard antenatal care [21]. Further training of the maternity care personnel and allocating resources to dietary and physical activity counselling in maternity care could be beneficial. However, changing the diet and physical activity habits in an effective way is challenging and new means alongside counselling might be needed to prevent the development of GDM [3]. Exploitation of digital approaches alongside standard care could prove to be feasible as these have shown promising results in the management of GDM [22].

We detected a lower dietary quality score in women with overweight/obesity compared to women with normal weight or underweight during pregnancy. This finding is in accordance with two previous American reports [13,23] and the Norwegian Mother and Child Cohort Study [24], which however used a different approach for the evaluation of the dietary quality score, i.e., calculation based on food frequency questionnaires, whilst our index was based on a specifically developed and validated stand-alone questionnaire to assess the overall diet with respect to that recommended [14]. In a recent study of women living with obesity, intakes of both macro- and micronutrients during pregnancy were found to be suboptimal [25]. As expected, the women with overweight/obesity in our study had a lower physical activity level and a lower level of education than the women with normal weight and they were more likely to have been smokers before they became pregnant, which suggests that not only a low dietary quality, but many other adverse lifestyle factors coalesce in women living with overweight or obesity, calling for new means to improve lifestyle habits in this at-risk group of women. This is of importance also due to the health risks that extend beyond pregnancy, including a heightened risk of type 2 diabetes in women with GDM [6]. Indeed, improving the dietary quality during pregnancy has been shown to be associated with a lowered weight gain up to eight years postpartum [26]. Based on our subgroup analysis, a normal or underweight pre-pregnancy BMI seemed to be a stronger indicator of good dietary habits than a history of GDM. This could be partly explained by the higher educational status of the normal-weight women in comparison to the women with overweight/obesity. Studies with a larger number of participants with prior GDM are required to clarify this observation.

In our study, only about every second woman met the recommended 150 min of physical activity weekly during pregnancy [27]; this value represented the category of moderate LTPA. This was slightly more than in the previous studies where the fulfilment of the recommendation was observed in less than every third woman in the first trimester of pregnancy [28] and in every fifth woman between weeks 10 and 24 of gestation [29]. Similar to our results, higher physical activity levels during pregnancy have been detected in women with normal weight as compared to women with overweight/obesity [28]. Regular exercise is known to improve blood glucose control [30] and could therefore be an important focus point for intervention strategies. In women with obesity, a higher level of physical activity has been shown to associate with better glycemic control five years postpartum [31].

The study population characteristics (Table 1) were close to the values reported in the Finnish perinatal statistics [4], e.g., the mean age of parturients (30.9 years in the perinatal statistics), the mean pre-pregnancy BMI (24.8 kg/m^2^), marital status (54% married), and the home region of the participants (Appendix A). These data indicated that the study population was representative of pregnant women throughout Finland, except that primiparous women were somewhat over-represented (41% in the perinatal statistics) and there was an under-representation of women who smoked during pregnancy (13% in the perinatal statistics) in the study. In addition, compared to Finnish women aged 15 or over [32], the study population had more likely a university degree (48% in the official education statistics).

The strength of this study was that it examined a representative sample of pregnant women from different parts of Finland, with the exception of the higher level of education of the participants, a typical characteristic of survey studies. Other strengths were its large sample size and its use of a validated stand-alone index to measure dietary quality. We also examined possible confounding factors. We recruited participants through social media, which is a recruitment method that fits modern lifestyles, especially pregnant women in the younger age groups [33] while this approach has also been shown to be an effective recruitment strategy in observational studies [34]. Self-reporting of the data may have introduced some inaccuracies. In addition, we did not inquire or exclude multiple pregnancies which might slightly influence the results, although the proportion of multiple pregnancies is only 1% in the general population [4], and thus also likely to have been similar in our participants. The IDQ was originally validated in non-pregnant adults [14] and the LTPA questionnaire uses terms like “becoming breathless”, which might be difficult to use as a marker in the classification of the intensity of physical activity in a pregnant woman. We did not have information on the dietary and physical activity habits of the women with a history of GDM during their previous pregnancies so we could not evaluate whether these had improved or deteriorated in any way. Future prospective studies should focus on evaluating this topic.

## 5. Conclusions

Many pregnant women have a low dietary quality and a light physical activity level in early to mid-pregnancy, this being especially pronounced in women with overweight or obesity. Having a history of GDM is not reflected as a good dietary quality or high physical activity level during pregnancy. New effective intervention strategies should be targeted at pregnant women, particularly women with overweight or obesity and women at risk for recurrent GDM, as these are likely to benefit from more intensive dietary and physical activity counselling from maternity care nurses or any new means to promote healthy lifestyle changes.

## Figures and Tables

**Table 1 nutrients-14-00651-t001:** Clinical characteristics of the women subdivided by history of gestational diabetes or their pre-pregnancy BMI values.

	All Women	Women with No History of GDM	Women with a History of GDM	Women with Normal Weight	Women with OverWeight/Obesity		
	*n* = 1034	*n* = 383	*n* = 86	*n* = 656	*n* = 378	*p*-Value ^a^	*p*-Value ^b^
Age (years), mean (SD)	29.4 (4.0)	30.3 (3.8)	30.8 (4.3)	29.2 (3.8)	29.7 (4.2)	0.35 ^c^	0.077 ^c^
Pre-pregnancy BMI (kg/m²), mean (SD)	24.8 (4.9)	24.5 (4.5)	28.0 (6.5)	21.9 (1.8)	29.9 (4.4)	<0.001 ^c^	<0.001 ^c^
Parity, *n* (%)						0.97 ^d^	0.15 ^d^
0	560 (54)	0 (0)	0 (0)	371 (57)	189 (50)		
1	315 (31)	257 (67)	58 (67)	190 (29)	125 (33)		
2	102 (10)	84 (22)	18 (21)	65 (10)	37 (10)		
≥3	52 (5)	42 (11)	10 (12)	28 (4)	24 (6)		
Marital status, *n* (%)						0.73 ^e^	0.92 ^d^
Married	518 (50)	230 (60)	52 (61)	331 (51)	187 (50)		
Cohabiting	474 (46)	143 (37)	31 (36)	301 (46)	173 (46)		
Single	27 (3)	4 (1)	2 (2)	16 (2)	11 (3)		
Other	14 (1)	6 (12)	1 (1)	8 (1)	6 (2)		
University degree, *n* (%)	690 (67)	229 (60)	56 (65)	456 (70)	234 (62)	0.39 ^e^	0.014 ^e^
Low income, *n* (%)	82 (8)	28 (7)	8 (9)	49 (8)	33 (9)	0.50 ^e^	0.55 ^e^
Smoking status, *n* (%)							
Smoked before pregnancy	169 (16)	50 (13)	20 (23)	84 (13)	85 (23)	0.028 ^e^	<0.001 ^e^
Smoked during pregnancy	23 (2)	9 (2)	3 (4)	13 (2)	10 (3)	0.46 ^e^	0.52 ^e^
Chronic disease ^f^, *n* (%)	120 (12)	43 (11)	14 (16)	70 (11)	50 (13)	0.20 ^e^	0.23 ^e^
Special diet ^g^, *n* (%)	276 (27)	97 (25)	26 (30)	182 (28)	94 (25)	0.35 ^e^	0.34 ^e^

GDM, gestational diabetes; SD, standard deviation; BMI, body mass index; ^a^ between history of GDM groups; ^b^ between normal-weight group and group with overweight/obesity; ^c^ Independent samples T-test; ^d^ Chi-square test; ^e^ Fisher’s exact test; ^f^ Type 1 diabetes (0.3%), type 2 diabetes (0.1%), cardiovascular disease (1.6%), coeliac disease (1.9%), irritable bowel syndrome (6.9%), or inflammatory bowel disease (1.3%); ^g^ Lactose-free (4.8%), milk-free (0.7%), gluten-free (3.1%), vegetarian (4.4%), low fermentable oligo-, di-, monosaccharides and polyols (low FODMAP, 1.2%), or several of these (12%) diets.

**Table 2 nutrients-14-00651-t002:** Dietary quality and physical activity in the women without and those with a history of gestational diabetes.

	All Women	Women with No History of GDM	Women with a History of GDM		
	*n* = 1034	*n* = 383	*n* = 86	*p*-Value ^a^	*p*-Value ^b^
IDQ score, mean (SD), adjusted mean (95% CI)	9.3 (2.2)	9.3 (9.1, 9.5)	9.5 (9.0, 9.9)	0.68 ^c^	0.61 ^d^
Good dietary quality, *n* (%)	479 (47)	173 (45)	38 (46)	1.0 ^e^	
MET-index (MET h/wk), median (IQR)	7.5 (12.0)	4.8 (10.1)	4.8 (10.1)	0.58 ^f^	
Categorized MET-index, *n* (%)				0.75 ^g^	
Light LTPA	460 (45)	198 (52)	44 (52)		
Moderate LTPA	450 (44)	146 (38)	35 (41)		
Vigorous LTPA	117 (11)	36 (10)	6 (7)		

IDQ, Index of Diet Quality; CI, confidence interval; MET, metabolic equivalent; IQR, interquartile range; LTPA, leisure-time physical activity. ^a^ between history of GDM groups, unadjusted; ^b^ between history of GDM groups, adjusted for pre-pregnancy BMI; ^c^ Independent samples T-test; ^d^ ANCOVA, adjusted for pre-pregnancy BMI; ^e^ Fisher’s exact test; ^f^ Mann–Whitney U-test; ^g^ Chi-square test.

**Table 3 nutrients-14-00651-t003:** Adherence to dietary recommendations in all the women, in the women without and those with a history of gestational diabetes and in the women with normal weight and women with overweight/obesity.

	All Women	Women with No History of GDM	Women with a History of GDM	Women with Normal Weight	Women with OverWeight/Obesity		
	*n* = 1034	*n* = 383	*n* = 86	*n* = 656	*n* = 378		
	*n* (%)	*n* (%)	*n* (%)	*n* (%)	*n* (%)	*p*-Value ^a^	*p*-Value ^b^
Vegetables daily	709 (69)	262 (69)	63 (76)	466 (72)	243 (65)	0.23 ^c^	0.017 ^c^
Fruit or berries daily	554 (54)	197 (52)	46 (55)	383 (59)	171 (46)	0.55 ^c^	<0.001 ^c^
Vegetables, fruit or berries ≥5 portions daily	436 (43)	157 (41)	37 (45)	292 (45)	144 (38)	0.62 ^c^	0.042 ^c^
Whole-grain products daily	610 (60)	243 (64)	44 (53)	425 (66)	185 (49)	0.062 ^c^	<0.001 ^c^
Vegetable oil -based spread on bread	331 (32)	129 (34)	27 (33)	212 (33)	119 (32)	0.90 ^c^	0.78 ^c^
Fish ≥2 portions weekly	291 (28)	107 (28)	18 (22)	196 (30)	95 (25)	0.28 ^c^	0.098 ^c^
Regular meal pattern	918 (90)	335 (88)	77 (93)	602 (93)	316 (84)	0.25 ^c^	<0.001 ^c^

^a^ between history of GDM groups; ^b^ between normal-weight group and group with overweight/obesity; ^c^ Fisher’s exact test.

**Table 4 nutrients-14-00651-t004:** Dietary quality and physical activity in the women with normal weight and in the women with overweight/obesity.

	Women with Normal Weight	Women with OverWeight/Obesity		
	*n* = 656	*n* = 378	*p*-Value ^a^	*p*-Value ^b^
IDQ score, adjusted mean (95% CI)	9.6 (9.5, 9.8)	8.8 (8.6, 9.1)	<0.001 ^c^	<0.001^d^
Good dietary quality, *n* (%)	341 (53)	138 (37)	<0.001 ^e^	
MET-index (MET h/wk), median (IQR)	7.5 (15.8)	4.8 (10.8)	<0.001 ^f^	
Categorized MET-index, *n* (%)			<0.001 ^g^	
Light LTPA	255 (39)	205 (54)		
Moderate LTPA	309 (48)	141 (37)		
Vigorous LTPA	86 (13)	31 (8)		

^a^ unadjusted; ^b^ adjusted for age; ^c^ Independent samples T-test; ^d^ ANCOVA, adjusted for age; ^e^ Fisher’s exact test; ^f^ Mann–Whitney U-test; ^g^ Chi-square test.

## Data Availability

The data presented in this study are available upon reasonable request from the corresponding author. The data are not publicly available due to the fact that they contain information that could compromise the privacy of the research participants.

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
