# Peer review of "Living with Overweight, Rather than a History of Gestational Diabetes, Influences Dietary Quality and Physical Activity during Pregnancy"

_nutrients, 2022, doi:10.3390/nu14030651_

Round 1

Reviewer 1 Report

In this study the authors evaluated the dietary quality and physical in pregnant women with or without previous GDM, and in overweight and obese women as compared to normal-weight women during pregnancy. The work is well designed and conducted. The sample size and the follow-up time are appreciable. The results are interesting. However, the authors must precise how GDM diagnosis has been made (criteria, timing, methodology). Is this diagnosis self-reported?

Author Response

Thank you for your comments. The diagnosis of gestational diabetes is self-reported. Based on the Current Care guidelines in Finland, nearly all pregnant women undergo a 2-hour oral glucose tolerance test (OGTT) in maternity care between gestational weeks 24 and 28. The exceptions when an OGTT is not routinely done are a primiparous woman under 25 years old with normal prepregnancy BMI and no family history of type 2 diabetes and a multiparous woman under 40 years old with normal prepregnancy BMI and no prior gestational diabetes or macrosomia of a newborn. If a pregnant woman has risk factors for gestational diabetes, such as BMI≥35, prior gestational diabetes, glycosuria in early pregnancy, family history of type 2 diabetes, corticosteroid medication or polycystic ovary syndrome, a 2-hour OGTT is done also between gestational weeks 12 and 16. A diagnosis of gestational diabetes is made if blood glucose is at or above the threshold levels in one or more timepoints: 0 h 5.3 mmol/l, 1 h 10.0 mmol/l, 2 h 8.6 mmol/l.

Lines 81-84 it was clarified that “The diagnosis of prior GDM was self-reported. Based on the Finnish guidelines [10], nearly all pregnant women undergo a 2-hour oral glucose tolerance test between gestational weeks 24 and 28. A diagnosis is made if blood glucose is at or above the threshold levels in one or more timepoints: 0 h 5.3 mmol/l, 1 h 10.0 mmol/l, 2 h 8.6 mmol/l.”

Reviewer 2 Report

Thank you for asking me to review this interesting paper; I have some suggestions that may help to improve clarity:

Title

The English is rather awkward here - I would suggest changing this to "Living with overweight, rather than history of gestational diabetes, influences diet and physical activity during pregnancy".

Abstract

Line 11 - change overweightedness to 'living with overweight'.

Can you add that overweight status was determined by BMI category. 

Line 22- particularly if overweight/living with obesity?

Introduction

Please define what you mean by 'Diet Quality' at the start of this introduction.

Can you also indicate what interventions are currently in place for women who are diagnosed with GDM; i.e. who should give the advice? when? what type of advice should be offered? Is there any indication of these interventions being effective for controlling GDM in the literature?

I strongly suggest that you use 'people first' language throughout this paper, particularly when describing obesity (e.g. 'Women living with obesity' rather than 'Obese women'). See ASO guidelines at https://aso.org.uk/sites/default/files/resources/2021-03/ASO-weight-stigma-and-discrimination-position-statement.pdf - for details.

Methods

Why was gestation < 28 weeks an inclusion criteria? Please explain.

Were there any further exclusion criteria? e.g. twin/multiple pregnancy; having a dietary restriction; having other pregnancy complications? Please expand on this.

Please can you also expand on the use of social media for recruitment; which sites were used & why?

There needs to be more details regarding height & weight data: was this taken from the medical notes (& so actually measured by a health professional), or was it self-reported on the questionnaire? If you are using self-reported height & weight some discussion of the limitations of using this is needed here (& in the limitations); did you introduce any methods to improve the accuracy of the height & weight data? If so, please indicate this here.

Discussion

The opening sentence is rather awkward; I suggest changing it to "Only half of Finnish women..."

As you grouped underweight & normal weight women together, please can you include underweight (with normal weight) in the discussion too.

The discussion focuses on the fact that women with a previous diagnosis of GDM, will have already received diet & PA advice in a previous pregnancy; so we would expect this to have influenced their current lifestyle, but it appears to have not. I would like to see further discussion of the quality and relevance of advice that women with GDM receive in Finland (both here and in the introduction). Who gives the advice? where? when? Does this advice need to be improved if it appears to have little effect on behaviour change? Can you make some further recommendations as to how interventions for GDM could be improved in future to have greater impact?

You may like to compare your findings to this recent paper: Charnley, M., Newson, L., Weeks, A. and Abayomi, J., 2021. Pregnant Women Living with Obesity: A Cross-Sectional Observational Study of Dietary Quality and Pregnancy Outcomes. Nutrients13(5), p.1652.

Author Response

Thank you for revising our article and for the constructive comments. We have made the modifications considering the advice.

Title

The English is rather awkward here - I would suggest changing this to "Living with overweight, rather than history of gestational diabetes, influences diet and physical activity during pregnancy".

R: Thank you for the suggestion, we changed the title to “Living with Overweight, rather than a History of Gestational Diabetes, Influences Diet Quality and Physical Activity during Pregnancy”

Abstract

Line 11 - change overweightedness to 'living with overweight'.

R: Line 12: "prepregnancy overweightedness” was changed to “living with overweight before pregnancy”.

Can you add that overweight status was determined by BMI category.

R: Line 17: The sentence was clarified “overweight status based on BMI category”.

Line 22- particularly if overweight/living with obesity?

R: Line 21: “Pregnant women, particularly those with overweight” was changed to “Pregnant women, particularly if living with overweight”.

Introduction

Please define what you mean by 'Diet Quality' at the start of this introduction.

R: Dietary quality was defined on lines 50-53: “A good dietary quality signifies an adherence to a health-promoting diet as defined in dietary recommendations with a high consumption of vegetables, fruits and berries, whole grain foods, vegetable oil-based spread and low-fat dairy and a low consumption of sugary beverages and food high in saturated fats.”

Can you also indicate what interventions are currently in place for women who are diagnosed with GDM; i.e. who should give the advice? when? what type of advice should be offered? Is there any indication of these interventions being effective for controlling GDM in the literature?

R: In Finland, the recommendation is that women with GDM receive dietary and physical activity counselling repeatedly from maternity care personnel, primarily nurses, in communal health centres. The aims of the counselling are maintaining normoglycemia and preventing excessive gestational weight gain as well as preventing the recurrence of GDM. The counselling should be individual and practical, based on following the Nordic nutrition recommendations. It has been suggested that in up to 70 to 85 % of cases, GDM can be controlled with lifestyle modifications (American Diabetes Association. 13. Management of Diabetes in Pregnancy. Diabetes Care, 2017, 40(Suppl 1), S114‒S119), but the counselling practises in the communal health centres have not been studied to large extent. Survey studies for maternity care nurses imply that the level of general nutrition knowledge of the maternity care personnel varies (Huurre, A.; Laitinen, K.; Hoppu, U.; Isolauri, E. How practice meets guidelines: Evaluation of nutrition counselling in Finnish well-women and well-baby clinics. Acta Paediatr 2006, 95, 1353‒1359), which we have pointed out in the Discussion section.

Lines 41-45: “In line with the current care guidelines, women with GDM receive diet and exercise counselling during pregnancy” was specified “In line with the current care guidelines [10], women with GDM receive diet and exercise counselling from maternity care nurses in communal health centres. The aims of the counselling are maintaining normoglycaemia and preventing excessive gestational weight gain by providing practical advice. In most cases, GDM can be controlled with lifestyle modifications alone [11].”

I strongly suggest that you use 'people first' language throughout this paper, particularly when describing obesity (e.g. 'Women living with obesity' rather than 'Obese women'). See ASO guidelines at https://aso.org.uk/sites/default/files/resources/2021-03/ASO-weight-stigma-and-discrimination-position-statement.pdf - for details.

R: Thank you for the suggestion. The language describing overweight status was changed throughout the manuscript to “women with overweight/obesity” or “women living with overweight/obesity”.

Methods

Why was gestation < 28 weeks an inclusion criteria? Please explain.

R: We wanted to recruit a comprehensive sample of women in early to mid-pregnancy. Considering that this is baseline data from a larger study in which women used a mobile app during pregnancy, we wanted to make sure that the women had sufficient time to use the mobile app before delivery. Therefore, we chose 28 weeks of gestation as the cut-off point. In addition, evaluating physical activity very late at pregnancy is more difficult because of likely restrictions due to physiological changes of the pregnancy and pregnancy complications.

It was added lines 70-72: “We wanted to recruit a cohort of women in early to mid-pregnancy, so the criteria for inclusion were less than gestational week 28 and Finnish language skills.”

Were there any further exclusion criteria? e.g. twin/multiple pregnancy; having a dietary restriction; having other pregnancy complications? Please expand on this.

R: We wanted to study a comprehensive sample of Finnish pregnant women, and therefore we did not have further exclusion criteria. Chronic diseases and special diets of the women studied are presented in Table 1 and did not differ between the study groups. We acknowledge that the inclusion of multiple pregnancies might affect the evaluation of diet and physical activity. Generally, similar foods are recommended for women with singleton and multiple pregnancies, but the recommended energy intake is slightly higher for women with multiple pregnancies.

Lines 300-303 it was added that “In addition, we did not inquire or exclude multiple pregnancies which might slightly influence the results, although the proportion of multiple pregnancies is only 1% in the general population [4], and thus also likely similar in our participants.”

Please can you also expand on the use of social media for recruitment; which sites were used & why?

R: Facebook was chosen as we had previously successfully used Facebook in recruiting pregnant women in another study. In Finland, most of the women in childbearing age have smartphones and most use social media, so recruiting women for an online study through social media seemed feasible.

Lines 67-69: ”Women were recruited into a study on lifestyle and the use of a mobile application during pregnancy by announcements in social media running from June to October 2017” was specified “Women were recruited into a study investigating lifestyle and the use of a mobile application during pregnancy by announcements in social media (Facebook) running from June to October 2017”.

There needs to be more details regarding height & weight data: was this taken from the medical notes (& so actually measured by a health professional), or was it self-reported on the questionnaire? If you are using self-reported height & weight some discussion of the limitations of using this is needed here (& in the limitations); did you introduce any methods to improve the accuracy of the height & weight data? If so, please indicate this here.

R: Because of the study design, the height and weight data of the participants were self-reported. Lines 78-79: “Data on dietary quality, physical activity, use of dietary supplements, current health status and weight were collected through electronic questionnaires” was specified “Data on dietary quality, physical activity, use of dietary supplements, current health status, height and weight were collected through electronic questionnaires”.

We are aware that self-reporting may introduce some inaccuracies, which we have pointed out in the Discussion-section lines 299 to 300. Weight during pregnancy is regularly measured by the maternity care nurses which likely increases the correctness of the self-reported weight data. The researchers calculated BMI from self-reported weight and height and checked for outliers in the data.

Discussion

The opening sentence is rather awkward; I suggest changing it to "Only half of Finnish women..."

R: Thank you for suggesting this. Lines 212-213: “We demonstrated that approximately only every second Finnish pregnant woman studied here” was changed to “We demonstrated that approximately only half of the Finnish pregnant women studied here”.

As you grouped underweight & normal weight women together, please can you include underweight (with normal weight) in the discussion too.

R: Thank you for noticing this. Lines 247-249: “We detected a lower dietary quality score in overweight/obese women compared to normal-weight women during pregnancy“ was changed to “We detected a lower dietary quality score in women with overweight/obesity compared to women with normal-weight or underweight during pregnancy”. Also, lines 265-267: “Based on our subgroup analysis, a normal prepregnancy BMI seemed to be a stronger indicator of good dietary habits than a history of GDM” was changed to “Based on our subgroup analysis, a normal or underweight prepregnancy BMI seemed to be a stronger indicator of good dietary habits than a history of GDM”.

The discussion focuses on the fact that women with a previous diagnosis of GDM, will have already received diet & PA advice in a previous pregnancy; so we would expect this to have influenced their current lifestyle, but it appears to have not. I would like to see further discussion of the quality and relevance of advice that women with GDM receive in Finland (both here and in the introduction). Who gives the advice? where? when? Does this advice need to be improved if it appears to have little effect on behaviour change? Can you make some further recommendations as to how interventions for GDM could be improved in future to have greater impact?

R: We added details on the standard diet and physical activity counselling for women with GDM in Finland in the Introduction section. The dietary and physical activity counselling for women with GDM is mostly given by maternity care nurses in the communal health centres. It has been studied that the counselling practices vary in Finland, as we have pointed out in the Discussion section lines 231 to 234. Based on this earlier research, further training of the personnel has been suggested to improve counselling. Interestingly, in a Finnish study, the risk of gestational diabetes was reduced by added dietary and physical activity counselling (above the usual practises) compared to the standard antenatal care. Allocation of resources to dietary and physical activity counselling in maternity care could therefore be beneficial. Further, addition of mHealth approaches alongside standard care could be beneficial as digital interventions have shown promising results in the management of GDM, namely improved glycaemic control and reduced risk of both maternal and foetal complications (Weihua, X.; Pinyuan, D.; Yu, Q.; Ming, W.; Bingquan, Y.; Xiaojin, Y. Effectiveness of telemedicine for pregnant women with gestational diabetes mellitus: an updated meta-analysis of 32 randomized controlled trials with trial sequential analysis. BMC Pregnancy Childbirth 2020, 20, 198).

Lines 234-237 it was added that “It is recommended that the dietary counselling for women with GDM should be tailored to the individual, practical and given repeatedly [10], but it is possible that these are not sufficiently implemented due to limited resources.” Lines 237-242: “Increasing healthy eating and physical activity may manifest as clinically improved health as demonstrated in a trial where the risk of GDM was reduced by an intervention which combined dietary and physical activity counselling [19]” was specified “Increasing healthy eating and physical activity may manifest as clinically improved health as demonstrated in a trial where the risk of GDM was reduced by combined dietary and physical activity counselling given above the standard antenatal care [21]. Further training of the maternity care personnel and allocating resources to dietary and physical activity counselling in maternity care could be beneficial.” Lines 244-246 it was added that: “Exploitation of digital approaches alongside standard care could prove to be feasible as these have shown promising results in the management of GDM [22].”

You may like to compare your findings to this recent paper: Charnley, M., Newson, L., Weeks, A. and Abayomi, J., 2021. Pregnant Women Living with Obesity: A Cross-Sectional Observational Study of Dietary Quality and Pregnancy Outcomes. Nutrients13(5), p.1652.

R: Thank you. We added lines 254-255: “In a recent study of women living with obesity, intakes of both macro- and micronutrients during pregnancy were found to be suboptimal [25].”

The modified article has been revised for English language by a native speaker.